# COVID-19 Preventive Behaviours in Cameroon: A Six-Month Online National Survey

**DOI:** 10.3390/ijerph18052554

**Published:** 2021-03-04

**Authors:** Joseph Nelson Siewe Fodjo, Leonard Ngarka, Wepnyu Y. Njamnshi, Leonard N. Nfor, Michel K. Mengnjo, Edwige L. Mendo, Samuel A. Angwafor, Jonas Guy Atchou Basseguin, Cyrille Nkouonlack, Edith N. Njit, Nene Ahidjo, Eric Samuel Chokote, Fidèle Dema, Julius Y. Fonsah, Godwin Y. Tatah, Nancy Palmer, Paul F. Seke Etet, Dennis Palmer, Dickson S. Nsagha, Daniel E. Etya’ale, Stephen Perrig, Roman Sztajzel, Jean-Marie Annoni, Anne-Cécile Zoung-Kanyi Bissek, Rose G. F. Leke, Marie-Thérèse Abena Ondoa Obama, John N. Nkengasong, Robert Colebunders, Alfred K. Njamnshi

**Affiliations:** 1Global Health Institute, University of Antwerp, 2610 Antwerp, Belgium; josephnelson.siewefodjo@uantwerpen.be (J.N.S.F.); robert.colebunders@uantwerpen.be (R.C.); 2Brain Research Africa Initiative (BRAIN), Yaoundé, Cameroon; lngarka@yahoo.com (L.N.); princeyembe@gmail.com (W.Y.N.); nfor.leonard@gmail.com (L.N.N.); mengnjomichel@yahoo.com (M.K.M.); edwigelaure.mendo@yahoo.fr (E.L.M.); angwaa@yahoo.co.uk (S.A.A.); jonas_guy82@yahoo.fr (J.G.A.B.); nkouonlack@gmail.com (C.N.); njitedith@yahoo.com (E.N.N.); neneolua@yahoo.com (N.A.); samuelchokote2006@gmail.com (E.S.C.); demfidel@yahoo.fr (F.D.); yundze.fonsah@gmail.com (J.Y.F.); tatahgodwin@gmail.com (G.Y.T.); palmernancylea@gmail.com (N.P.); paul.seke@gmail.com (P.F.S.E.); palmerdd47@gmail.com (D.P.); dsnsagha@gmail.com (D.S.N.); roseleke@yahoo.com (R.G.F.L.); mtaobama@yahoo.fr (M.-T.A.O.O.); 3Brain Research Africa Initiative (BRAIN), 1226 Geneva, Switzerland; etyaaled@gmail.com (D.E.E.); Stephen.Perrig@hcuge.ch (S.P.); Roman.Sztajzel@hcuge.ch (R.S.); Jean-Marie.annoni@unifr.ch (J.-M.A.); 4Division of Health Operations Research, Ministry of Public Health, Yaoundé, Cameroon; annezkbissek@yahoo.fr; 5CDC Africa, African Union, Addis Ababa 3243, Ethiopia; nkengasongJ@africa-union.org

**Keywords:** COVID-19, Cameroon, preventive measures, adherence, survey

## Abstract

Since March 2020, the Cameroonian government implemented nationwide measures to stall COVID-19 transmission. However, little is known about how well these unprecedented measures are being observed as the pandemic evolves. We conducted a six-month online survey to assess the preventive behaviour of Cameroonian adults during the COVID-19 outbreak. A five-point adherence score was constructed based on self-reported observance of the following preventive measures: physical distancing, face mask use, hand hygiene, not touching one’s face, and covering the mouth when coughing or sneezing. Predictors of adherence were investigated using ordinal logistic regression models. Of the 7381 responses received from all ten regions, 73.3% were from male respondents and overall mean age was 32.8 ± 10.8 years. Overall mean adherence score was 3.96 ± 1.11 on a scale of 0–5. Mean weekly adherence scores were initially high, but gradually decreased over time accompanied by increasing incidence of COVID-19 during the last study weeks. Predictors for higher adherence included higher age, receiving COVID-19 information from health personnel, and agreeing with the necessity of lockdown measures. Meanwhile, experiencing flu-like symptoms was associated with poor adherence. Continuous observance of preventive measures should be encouraged among Cameroonians in the medium- to long-term to avoid a resurgence in COVID-19 infections.

## 1. Introduction

The coronavirus disease 2019 (COVID-19) has been a major global health concern in 2020, as a stunning 81,484,663 cases and 1,798,160 deaths were cumulatively reported around the world as of 31 December 2020 [1]. The dynamics and evolution of the disease vary widely from one context to another, with developed nations recording the highest death toll. Despite very pessimistic COVID-19 predictions for countries in sub-Saharan Africa, it appears that they have been least affected by the pandemic, as this part of the world accounts for barely 5% of the global COVID-19 burden [2]. Among other reasons, it is speculated that under-reporting of cases, early effective lockdown measures, young age of the population, previous exposure to other coronaviruses, a strong immune system because of frequent exposure to pathogens, and geographical and/or genetic factors contributed to mitigating the health impacts of COVID-19 in Africa [3,4]. The role of traditional medicine and particularly the African medicinal plants [5,6] in limiting COVID-19 transmission deserves further research in the African context. 

In Cameroon, the first case of COVID-19 was identified on 6 March 2020, when a French national returning from Europe tested positive for the severe acute respiratory syndrome coronavirus-2 (SARS-CoV-2). Thereafter, a COVID-19 task force was formed, and international borders were closed for incoming passengers by 18 March 2020. Under the leadership of the Prime Minister, several preventive measures were instituted nationally to contain the local COVID-19 outbreak. These included: closure of all schools and training institutions, forbiddance of any gathering of more than fifty persons, closure of entertainment spots by 18:00 daily, strong discouragement of urban and inter-urban travel, consumer flow to be regulated in markets and shopping centres, postponement of sports competitions, and observance of hygiene measures such as regular hand washing with soap, avoiding close contacts with other persons, covering one’s mouth when coughing/sneezing, and other measures as prescribed by the World Health Organization (WHO) [7]. As the epidemic continued gaining ground, the national COVID-19 response equally ramped up with the institution of additional measures, including mandatory face mask use in public places, effective from 13 April 2020 [8]. In addition, contact tracing strategies were intensified and testing capacity also increased to over 1000 tests per day in May 2020. Furthermore, a toll-free phone line (1510) was dedicated to the COVID-19 response team, through which the population could obtain support or forward any COVID-19 related information. Despite all these coordinated actions by the government, the number of confirmed COVID-19 cases increased, with a peak number of new cases per day reaching 1445 on 6 July 2020 [9]. Little is known regarding the population’s adherence to the prescribed measures in such an unprecedented context of nationwide restrictions for public health reasons and such knowledge is essential for government policy concerning the current outbreak as well as future pandemics. Therefore, this study aimed at obtaining feedback from the population to understand their experience of the first ever national lockdown and to identify determinants of adherence to the COVID-19 preventive measures in Cameroon. 

## 2. Methods 

### 2.1. Study Setting and Population

The study was conducted from 5 June to 5 December 2020 all over the national territory of Cameroon, a country located in the Central African sub-region. Administratively, the country counts ten regions and uses two official languages: French and English. The median age of the Cameroonian population is estimated at 18.7 years, with a greater proportion (56.3%) of residents living in urban settings [10]. The COVID-19 outbreak occurred while Cameroon was already faced with a 4-year civil crisis in the two English-speaking regions (Northwest and Southwest) and security concerns in the northern and eastern parts of the country. 

### 2.2. Study Tool and Procedures

We conducted an online survey designed through a collaboration of the Brain Research Africa Initiative (BRAIN) with the International Citizen Project on COVID-19 (ICPCovid) consortium [11]. The ICPCovid website, initiated in March 2020 by a team of researchers based at the University of Antwerp in Belgium, offers a secure electronic platform to collect COVID-19-related data from several low- and middle-income countries in order to assess adherence to preventive measures. For this study, the ICPCovid template questionnaire was adapted to the Cameroonian context by BRAIN investigators and translated into French and English. Questions regarding adherence to specific measures, as well as experience of flu-like symptoms during the 14 days preceding participation, were asked in a “yes/no” format. Furthermore, we asked respondents to quantify the difficulty they personally encountered in observing the stay-at-home instructions issued by the government using a 5-point Likert scale (ranging from 1 = not difficult at all to 5 = extremely difficult). The web-link to the electronic survey was disseminated via social media platforms and by repeated bulk messaging to phone users and media sensitisation facilitated by the Ministries of Communication and Post & Telecommunications and their partner agencies. Upon clicking on the link, the user was directed to an information and consent page where he/she could agree to participate, fill in the responses, and submit them via a smartphone, tablet, or computer. The electronic questionnaire was continuously open to receive responses, except for short periods when it was temporarily closed for maintenance and review. All submitted responses were time-stamped and immediately stored in a password-protected server in Belgium until data retrieval. We did not calculate a sample size for this study, but rather chose the convenience approach whereby all eligible responses received during the study period would be analysed.

### 2.3. Data Analysis

Collected data were exported to Microsoft Excel 2016 spreadsheets for cleaning and later transferred to R version 4.0.2 for analysis. Continuous variables were summarized as means and standard deviation (SD), while categorical variables were expressed as percentages. A 5-point adherence score was constructed based on the respondents’ reported observance of the following preventive measures: mask use, physical distancing, hand hygiene (regular hand washing with soap and/or use of alcohol-based hand gel), not touching one’s face, and covering the mouth when coughing/sneezing. Observance of each measure scored one point, and otherwise zero; the final score ranged from 0–5 with a higher score implying higher adherence to the preventive measures. In case of missing values for any of the above-mentioned preventive measures, the ‘partial’ adherence scores were standardized to a 5-point scale using the formula below and rounded to the nearest whole number: $$Adjusted\;Score\;\left(on\;5\right)=\frac{Total\;score\;for\;“n”\;preventive\;measures}{n}\times 5$$

In order to account for the timing of the responses when assessing preventive behaviours, we grouped the respondents based on the week of participation, and investigated a possible association between the mean adherence score during a given week *n* and the number of new reported cases in Cameroon [9] during the following week *n* + 1. This approach was chosen because the SARS-Cov-2 has a mean incubation period of 5.8 days [12], and thus it would take about a week for any change in adherence to translate into a change in COVID-19 incidence. Only weeks with ≥100 responses were analysed when constituting the weekly clusters. Socio-demographic determinants of adherence to COVID-19 preventive measures were investigated by constructing an ordinal logistic regression model with clustered standard errors, using the 5-point adherence score as the dependent variable. The model was done using the *polr* function (package: ‘MASS’) and the *vcovCL* function (package: ‘sandwich’) in the software R. Survey weights were applied based on the number of participants per week using inversed probability weighting. The overall weekly weight for week *n* was given by: total survey population divided by the number respondents during week *n*. We further divided this weekly weight by the number respondents during week *n* to obtain individual weighting coefficients. Covariates for the final model were selected based on a *p*-value < 0.2 in univariate analysis. All *p*-values < 0.05 were considered statistically significant. 

### 2.4. Ethical Considerations

This study was approved by the National Ethics Committee of Cameroon (Ref: 2020/05/1229/CE/CNERSH/SP of 06.5.20) as well as the Ethics Committee of the University of Antwerp, Belgium (Ref: 20/13/148). Only data from participants aged 18 years and above who provided an e-consent were retained for analysis. All data were collected anonymously and treated with absolute confidentiality.

## 3. Results

### 3.1. Participant Characteristics

Of the 7538 responses received, 7381 were retained for analysis after data cleaning. The mean age of participants was 32.8 years (range: 18–89 years), and there were more male participants (Table 1). Majority of participants lived in urban settings (72.9%) and had attained a university level of education (70.9%). The least represented region in the study population was the East region with 181 participants, while the most represented regions were the Centre and Littoral regions, with 2193 and 1764 participants, respectively. 

### 3.2. COVID-19 Preventive Behaviours

The overall mean adherence score of participants was 3.96 ± 1.11 on a scale ranging from zero to five. More than four-fifths of the participants had an adherence score ≥3. The most observed measure for COVID-19 prevention was “not going to bars/restaurants during the past seven days”, which was reported by 96.8% of respondents. Adherence was significantly different by sex for the following measures: physical distancing, coughing hygiene, staying at home in the event of flu-like symptoms, and having been to a market during the past seven days (Table 2). Among the mask users (*n* = 6431), the most common mask type was the reusable cloth mask (71.7%), followed by surgical masks (24.0%) and filters such as N95, KN95, and FFP2 (4.3%). Among the non-mask users in our study (*n* = 858), the reasons given for not wearing face masks included financial limitations for 134 (15.6%), lack of knowledge about where to find masks for 14 (1.6%), complaints about masks being uncomfortable for 571 (66.6%), and the conviction that masks are unnecessary for 354 (41.3%). 

Mean adherence scores were similar across residential settings (rural vs. suburban vs. urban; *p* = 0.294). However, adherence scores were higher among females compared to males (mean scores 4.02 ± 1.08 vs. 3.94 ± 1.12; *p* = 0.003) and higher among respondents from the healthcare sector (4.03 ± 1.06 vs. 3.95 ± 1.12; *p* = 0.048). While the adherence score was significantly different across the ten regions of Cameroon, no disparity was observed in the reported difficulty to comply with the stay-at-home instructions of the government (Table 3). 

Considering only study weeks with more than 100 responses, the mean weekly adherence score during week *n* was not significantly associated with the COVID-19 incidence during week *n* + 1 (Pearson correlation coefficient = 0.166, *p* = 0.647). A graphical illustration of the evolution of the COVID-19 burden and adherence scores by week is shown in Figure 1. 

### 3.3. Flu-Like Symptoms Reported by Participants

Overall, 3338 (45.2%) participants reported experiencing at least one flu-like symptom during the two weeks preceding participation in the survey. Applying the WHO’s updated case definition for COVID-19 screening [13] revealed a prevalence of suspected COVID-19 cases of 1193/7381 (16.2%) in our survey population. Of note, myalgia and dry cough were more frequent among respondents from the healthcare sector, while all other symptoms had similar frequencies among persons involved in healthcare and other respondents. Among the respondents who were tested for COVID-19 (by PCR or antigen-detecting rapid tests) within the two weeks preceding survey participation (*n* = 496), those with positive test results (*n* = 20; positivity rate = 4.0%) reported more frequent flu-like symptoms during this period (Table 4).

The ordinal logistic regression model revealed that being older, being a male, having the opinion that lockdown measures are necessary to control COVID-19 in Cameroon, obtaining COVID-19 information from healthcare workers, and having been tested for COVID-19 at least once were associated with increased odds for higher adherence scores (Table 5). In contrast, being a healthcare worker/student, obtaining COVID-19 information from social media, or belonging to the middle social class reduced the odds for high adherence. 

## 4. Discussion

Our study shows that adherence to COVID-19 preventive measures in Cameroon was moderate to high initially. However, adherence started declining during the latter phase of the outbreak, from week 40 (Figure 1) after new cases of COVID-19 started decreasing steadily all over the national territory. The decrease in adherence during the latter weeks preceded the rebound incidence of COVID-19 cases observed towards the end of the study period. Our findings complement previous data about the evolution of adherence to COVID-19 preventive measures in Cameroon as reported during earlier periods of the outbreak. Surveys conducted in some parts of Cameroon between March and May 2020 [14,15,16] already found satisfactory levels of knowledge about COVID-19, and good adherence to preventive measures such as face mask use (100% adherence in May 2020 [16]). Overall, it appears that adherence increased steadily during the exponential phase of the outbreak (March to August 2020) probably due to fear, considering the devastating reports of COVID-19 morbidity and mortality in China, Europe, and the Americas. Upon realising that COVID-19 was much less damaging in Cameroon, adherence trends eventually took a downward trajectory. One possible reason to account for the relatively lower COVID-19 burden in Cameroon despite the decreasing adherence to preventive measures could be the early closure of international borders and implementation of nationwide restrictions as early as March 2020. Of note, caution must be exercised when interpreting the adherence rates reported in our study or previous online surveys in Cameroon, as they are most likely overestimations. Given the sampling bias associated with the web-based approach (respondents would mostly be educated persons with internet access and often residing in urban settings), our study population is likely to report satisfactory preventive behaviours, which may not necessarily reflect the overall attitudes of the general population.

We observed that adherence varied significantly across the different geographical regions of Cameroon. Regional disparities regarding the burden of COVID-19 in Cameroon had equally been reported by Judson et al. [17], who found that the Centre region (one of the regions where mean adherence scores were lowest) had the highest number of cases per 100,000 persons. Meanwhile, the COVID-19 burden was lowest in the Northwest region alongside the Northern, Eastern, and Southern parts of the country [17], where we found adherence scores to be higher. In the same light, we noted that experiencing flu-like symptoms was associated with lower adherence scores (Table 5). When taken together, these findings suggest that increasing adherence might lower the burden of COVID-19 and vice-versa; however, this trend was not very apparent in Figure 1, probably because the data was not presented by region. Detailed information about adherence to the individual preventive measures included in the five-point score and disaggregated by region are reported in the Appendix A and show significant geographical disparities for all five measures (Appendix A). One possible contributing factor for the highest adherence scores in the Northwest region is the ongoing civil unrest in that part of the country, which is more severe in that region and had caused several families to adopt a stay-at-home attitude even prior to the COVID-19 outbreak. In contrast, busy cities such as Yaoundé and Douala may have found it more difficult to quickly adapt to the government-issued COVID-19 restrictions, thereby lowering the overall adherence scores in the Centre and Littoral regions, respectively. 

The multivariable analysis showed that older persons as well as those who had once been tested for COVID-19 (mostly contacts/exposed individuals) had higher odds of adhering to the preventive measures. These groups of respondents were more keen to adopt stringent COVID-19 preventive behaviours, most likely because they were more conscious that they stood a higher risk of becoming infected/severely ill. Similar to what was observed in Somalia [18], obtaining COVID-19 information from healthcare workers was significantly associated with higher adherence levels in Cameroon. In this regard, it is crucial that actors in the health sector be properly informed about COVID-19-related scientific advances, particularly regarding prevention. A recent report from the West region of Cameroon (lowest adherence scores) found that COVID-19 knowledge was poor even among healthcare workers [19], further underlining the urgency to establish up-to-date COVID-19 training platforms for the health personnel who will in turn educate the rest of the population regarding healthy attitudes and practices to minimize COVID-19 transmission. 

We found that men observed physical distancing less frequently than women (Table 2); this suggests that men were likely in close proximity with other persons outside their household, possibly for professional reasons and particularly in the informal sector. Paradoxically, being a male was associated with increased adherence (Table 5), suggesting that although men were seldom confined in their homes, they did their best to respect other preventive measures. In Cameroon, men are culturally considered as the breadwinners of the household and were therefore obliged to keep working in order to provide for their families despite the implemented restrictions. A survey conducted in rural Cameroon (West region) prior to the local COVID-19 outbreak in the country found that over 70% of participants resorted to the informal sector to provide for their families, and 85% of respondents reported their inability to comply to confinement measures if they were introduced [14]. These realities highlight the necessity of developing context-specific public health measures that would fit with different settings in order to achieve optimal compliance. For instance, from our findings, we can deduce that sensitisation about the importance of face masks in COVID-19 prevention, coupled with the provision of free/cheap cloth masks that are designed to be optimise the comfort of the user, would greatly boost mask use among Cameroonians. Also, educating people about the importance of lockdown measures to halt COVID-19 transmission would likely improve their adherence to the government instructions. Ultimately, the decentralization of the national COVID-19 response to allow different regions/municipalities to implement interventions they deem fit for their respective populations would be more fruitful [20]. Context-specific COVID-19 preventive strategies may need be implemented and sustained in Cameroon to avoid a possible resurgence of SARS-CoV-2 transmission, such as that reported recently in Kenya [21].

Flu symptoms were reported by 3338 (45.2%) of respondents, with 1193/3338 (35.7%) of them meeting the WHO definition for suspected COVID-19 cases. The fact that experiencing flu symptoms was negatively associated with adherence scores suggests that the low compliance of symptomatic respondents is what increased their chances of developing the reported symptoms. Furthermore, dry cough (a hallmark symptom of COVID-19) was more common among healthcare workers, suggesting that they may be more at risk to become infected compared to the general population. In the absence of COVID-19 test results, it remains unclear whether the respondents who met the clinical definition actually had the disease. Many more persons may have also been infected with the SARS-CoV-2 albeit being asymptomatic [22]. Continued compliance to preventive measures is therefore recommended as long as community transmission is ongoing, since COVID-19 spreaders could appear perfectly healthy in about half of cases [22]. In that light, mathematical modellers recommended that confinement (even partial), alongside mass masking and other hygienic rules, is indispensable to eradicate COVID-19 in Cameroon [23,24]. It could be worthwhile to investigate the effects of traditional medicine use on both the occurrence of flu-like symptoms among Cameroonians during the COVID-19 outbreak, and also on the preventive behaviours of avid consumers of home-made concoctions, as the latter may feel more protected against the virus than others. 

Our study is not devoid of limitations. Our respondents cannot be considered as a representative sample of the general population, since only literate persons with access to the internet were able to participate. According to the World Bank data, internet penetration in Cameroon was around 23% in 2019 [25], so our study may have recruited the minority of Cameroonians with possible higher socioeconomic status and these findings could not reflect the real adherence to preventive measures among Cameroonians. The fact that male respondents were over-represented may have biased some of the conclusions. We found two studies that analysed data of internet use in Cameroon by gender. In one study from 2010, 13.1% of men and 12.8% of women were reported to use the internet [26]. The second study, assessing internet adoption by men and women in 2015, reported that being male, higher education level, and being aged below 45 years were found to be determinants of internet use [27]. Also, the online approach makes it impossible to verify the reality of the situation in the field at the time of response. However, we are inclined to embrace the observed trend of declining adherence as shown by our data, because by the end of November 2020, less than half of the people encountered in the streets of Yaoundé had their face masks on (personal observation, J.N.S.F.). In view of a new rise in COVID-19 incidence at the end of the study period, it is imperative that Cameroonians adopt more stringent preventive behaviours to curb the current evolution of the COVID-19 burden. 

## 5. Conclusions

Adherence to COVID-19 preventive measures in Cameroon was initially high, but appeared to decline over time. It is vital that the population remains on its guard, as COVID-19 incidence was on the rise during the last study weeks. Context-specific COVID-19 preventive strategies should be implemented and sustained in Cameroon to avoid a possible resurgence of SARS-CoV-2 transmission. Continuous adherence to preventive measures should be encouraged among Cameroonians, because large-scale COVID-19 vaccination may not be rolled out in the near future.

## Figures and Tables

**Figure 1 ijerph-18-02554-f001:**
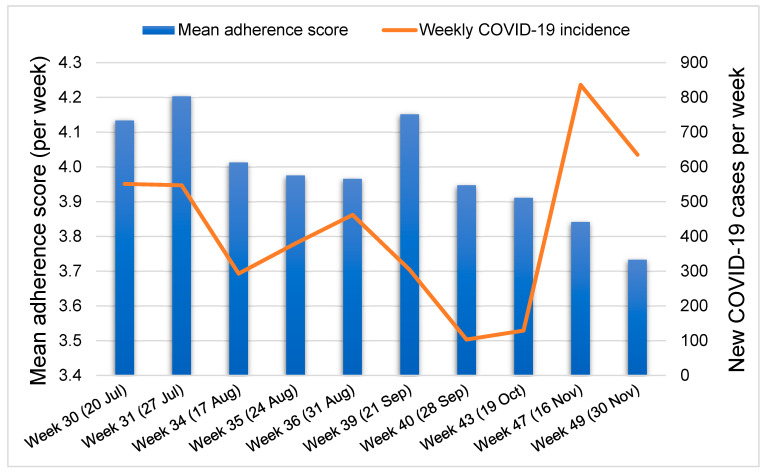
Evolution of COVID-19 burden and adherence scores in Cameroon.

**Table 1 ijerph-18-02554-t001:** Socio-demographic characteristics of respondents.

Characteristics	Survey Findings *N* = 7381
Age in years: Mean (SD)	32.8 (10.8)
Gender: *n* (%)	
Male	5409 (73.3%)
Female	1972 (26.7%)
Education level: *n* (%)	
Primary school	118 (1.6%)
Secondary school	2029 (27.5%)
University: undergraduate	2963 (40.1%)
University: postgraduate	2271 (30.8%)
Residential setting: *n* (%)	
Rural	672 (9.1%)
Suburban	1327 (18.0%)
Urban	5382 (72.9%)
Live alone in household: *n* (%)	1383 (18.7%)
Professional status: *n* (%)	
Student	1832 (24.8%)
Unemployed	1342 (18.2%)
Self-employed	869 (11.8%)
Private employee	1843 (25.0%)
Government employee	1285 (17.4%)
Retired	210 (2.9%)
Self-reported socio-economic status: *n* (%)	
Low	2494 (33.8%)
Lower-middle	3839 (52.0%)
Upper-middle	912 (12.4%)
High	136 (1.8%)
Healthcare worker or student: *n* (%)	793 (10.7%)
Source of COVID-19 information: *n* (%) ^a^	
Radio, television, or other government channels	6601 (89.4%)
Social media	5780 (78.3%)
Health personnel	2874 (38.9%)
Agree that lockdown is necessary for COVID-19 control in Cameroon	4638 (62.8%)
Underlying chronic disease: *n* (%) ^b^	716 (9.7%)
Tested for COVID-19: *n* (%)	2042 (27.7%)
Tested during the 2 weeks preceding participation in the survey: *n* (%)	521/2042 (25.5%)
COVID-19 test results for those tested last two weeks before responding to survey: *n* (%)	
Positive	20/496 (4.0%)
Negative	476/496 (96.0%)
Not known	25

^a^ Each participant was allowed to choose more than one answer, and hence the categories may overlap. ^b^ Heart disease, diabetes, hypertension, cancer, HIV, or asthma.

**Table 2 ijerph-18-02554-t002:** Reported COVID-19 preventive behaviours and adherence scores among participants.

Characteristics	Male *N* = 5409	Female *N* = 1972	*p*-Value
Wearing a face mask: *n* (%) ^a^	4691 (87.9%)	1740 (89.2%)	0.118
Observing 1.5 m physical distancing: *n* (%)	3664 (67.7%)	1252 (63.5%)	0.001
Regular handwashing: *n* (%)	4844 (89.6%)	1789 (90.7%)	0.154
Regular use of alcohol-based gel: *n* (%)	3621 (66.9%)	1320 (66.9%)	1.000
Covering mouth after coughing/sneezing: *n* (%)	4330 (80.1%)	1636 (83.0%)	0.005
Avoiding touching face (eyes, nose, mouth): *n* (%)	3635 (67.2%)	1343 (68.1%)	0.482
Staying home when I feel flu-like symptoms: *n* (%) ^b^	2158 (67.4%)	807 (70.8%)	0.037
Been to a bar/restaurant in the past 7 days: *n* (%)	187 (3.5%)	50 (2.5%)	0.056
Been to a market in the past 7 days: *n* (%)	3658 (67.6%)	1353 (68.6%)	0.046
Been to a religious gathering in the past 7 days: *n* (%)	731 (13.5%)	258 (13.1%)	0.658
Travelled during the past 7 days: *n* (%)	1583 (29.3%)	597 (30.3%)	0.417
**Adherence score: *n* (%)**			
0	15 (0.3%)	5 (0.3%)	0.022
1	165 (3.1%)	41 (2.1%)	
2	484 (9.0%)	174 (8.8%)	
3	973 (18.0%)	305 (15.5%)	
4	1602 (29.6%)	604 (30.6%)	
5	2170 (40.1%)	843 (42.7%)	

^a^ 92 missing values (70 among males, 22 among females); ^b^ excludes 3038 participants who reported to have never experienced symptoms (2206 males, 832 females).

**Table 3 ijerph-18-02554-t003:** Adherence and difficulty scores by region.

Region	Adherence Score (0–5)		Difficulty Likert Score (1–5)	
	Mean (SD)	*p*-Value *	Mean (SD)	*p*-Value *
Northwest	4.16 (1.03)	<0.001	3.34 (1.47)	0.741
Adamawa	4.08 (1.07)		3.20 (1.49)	
East	4.07 (0.97)		3.22 (1.34)	
Southwest	4.03 (1.13)		3.24 (1.51)	
South	4.01 (1.05)		3.20 (1.47)	
Far North	3.97 (1.06)		3.33 (1.53)	
North	3.94 (1.05)		3.35 (1.43)	
Littoral	3.92 (1.13)		3.31 (1.46)	
Centre	3.90 (1.15)		3.31 (1.35)	
West	3.89 (1.08)		3.27 (1.47)	

* Kruskal–Wallis test.

**Table 4 ijerph-18-02554-t004:** Flu-like symptoms and COVID-19 test results within the two weeks preceding participation in the survey.

Symptoms during the Past 14 Days	Positive COVID-19 Test: *n* (%) *N* = 20	Negative COVID-19 Test: *n* (%) *N* = 476	*p*-Value
Asymptomatic	6 (30.0%)	283 (59.5%)	0.017
Fever	10 (50.0%)	54 (11.3%)	<0.001
Headaches	8 (40.0%)	110 (23.1%)	0.105
Dry cough	4 (20.0%)	30 (6.3%)	<0.001
Productive cough	4 (20.0%)	18 (3.8%)	0.009
Sore throat	6 (30.0%)	17 (3.6%)	<0.001
Coryza	8 (40.0%)	47 (9.9%)	0.001
Loss of smell (anosmia)	7 (35.0%)	8 (1.7%)	<0.001
Loss of taste (ageusia)	6 (30.0%)	16 (3.4%)	<0.001
Shortness of breath	4 (20.0%)	5 (1.1%)	<0.001
Myalgia	7 (35.0%)	37 (7.8%)	0.001
Fatigue	8 (40.0%)	49 (10.3%)	0.001
Nausea	1 (5.0%)	12 (2.5%)	0.418
Diarrhoea	3 (15.0%)	20 (4.2%)	0.059
**Suspected/Probable COVID-19 Cases: *n* (%)**			
Based on the WHO updated definition [13]	10 (50.0%)	66 (13.9%)	<0.001

**Table 5 ijerph-18-02554-t005:** Ordinal logistic regression investigating factors associated with adherence score, with standard errors clustered according to week of participation.

Covariates	Adjusted OR	*p*-Value
	(95% CI)	
Age (in years)	1.005 (1.005–1.006)	<0.001
Male gender	1.242 (1.228–1.256)	<0.001
Region		
Centre	Reference	
Adamawa	1.106 (1.089–1.123)	<0.001
East	0.669 (0.660–0.679)	<0.001
Far North	0.905(0.897 –0.914)	<0.001
Littoral	1.032 (1.019–1.045)	<0.001
North	0.995 (0.986–1.003)	0.212
Northwest	1.389 (1.368–1.410)	<0.001
South	0.921 (0.901–0.941)	<0.001
Southwest	1.371 (1.361–1.381)	<0.001
West	0.654 (0.642–0.667)	<0.001
Living along in household	1.037 (1.028–1.045)	<0.001
Profession		
Student	Reference	
Jobless	0.766 (0.760–0.772)	<0.001
Self-employed	0.744 (0.737–0.751)	<0.001
Private worker	1.080 (1.069–1.091)	<0.001
Government worker	1.506 (1.490–1.522)	<0.001
Retired	0.824 (0.809–0.840)	<0.001
Healthcare worker or student	0.919 (0.914–0.925)	<0.001
Socio-economic status/class		
Low	Reference	
Lower-middle	0.821 (0.817–0.825)	<0.001
Upper-middle	0.548 (0.539–0.557)	<0.001
High	1.075 (1.047–1.102)	<0.001
Flu symptom(s) during past 14 days	0.526 (0.524–0.527)	<0.001
COVID-19 information from healthcare workers	1.057 (1.048–1.067)	<0.001
COVID-19 information from social media	0.920 (0.917–0.924)	<0.001
Agree that lockdown is necessary in Cameroon	2.117 (2.100–2.135)	<0.001
Has been tested for COVID-19	1.912 (1.895–1.929)	<0.001

OR: odds ratio; CI: confidence interval.

## Data Availability

The data presented in this paper are available upon reasonable request to the corresponding author.

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
