# Peer review of "COVID-19 Preventive Behaviours in Cameroon: A Six-Month Online National Survey"

_ijerph, 2021, doi:10.3390/ijerph18052554_

Round 1
Reviewer 1 Report
Major concerns:
i) The sample design is not representative. Sample size calculation was not included.
ii) As the data was collected weekly, there would be double counting problem. It indicates that same person can submit multiple questionnaires during the data collection period.
iii) Figure 1 indicates weekly adherence scores, which also supports the previous observation.
iv) Survey weight was not considered (unweighted n and weighted %).
v) Survey weight was not considered in regression model. Check the link: https://stats.idre.ucla.edu/stata/seminars/survey-data-analysis-with-stata-15/
Reviewer 2 Report
I congratulate Dr. Njamnshi and his team for doing this interesting study. This is a six-month online survey study to assess the preventive behavior of Cameroonian adults during the COVID-19 outbreak.
1) Line 41, 42: frequent recourse to traditional medicine, climatological.
There is no study to support the positive role of traditional medicine to prevent Covid-19.
Different studies ruled out the role of African climate in the incidence of Covid-19
2) Line 117,118,119: investigated a possible association between the mean adherence score per week and the number of new reported cases in Cameroon during the same week.
This association does not have clinical value due 3–5-day incubation period of Sars-Cov-2. To measure the consequence of any public health measures to manage the covid-19, we should analyze the epidemiology of Covid-19 after 1 or 2 weeks of implementing the policy. In this study, to assess the possible relationship between the mean adherence score and reported new cases of covid-19, we should analyze the number of newly reported cases in the next 2 weeks, not the same week. For example, if you have a high or low score of adherence score in the first week of May, the effect of this adherence will be revealed in the second and third weeks of May, not the same week. Please amend your analysis accordingly.
3) Line 145 – Table 1
There is a huge difference between the male (73%) and Female ( 27%) participants. What is the reason for this difference? Please add this issue and any possible explanation to the limitation part of the paper.
4) Line 145 – Table 1
Self-reported socio-economic status: n (%) :Low 2494 (33.8%), Lower-middle 3839 (52.0%), Upper-middle 912 (12.4%), High
Please explain how did you define Low to high socio-economic status? Is it based on income? Education?
5) Line: 169 -170-171: adherence score appeared to be inversely related to the weekly COVID-19 incidence in Cameroon, albeit non-significantly (Pearson correlation coefficient= –0.190, p=0.599).
Same week adherence score and COVID-19 incidence are not clinically relevant. It should be amended accordingly based on my explanations in comment 2.
6) Line 183-184: Among the respondents who were tested for COVID-19 within the two weeks preceding survey participation (n=496), those with positive test results (n=20; positivity rate = 4.0%) reported more frequent flu-like symptoms during this period.
Please specify the testing method. Was it serological or molecular?
7) Line188: being of the opinion
Change being of to having
8) Line 264: respondents already admitted that they will not be able to comply should confinement measures.
Please correct this sentence.
9) According to the world bank data, internet penetration in Cameroon was around 23% in 2019 so the data of this study represent the minority of Cameroonians with possible higher socioeconomic status and these findings could not reflect the real adherence to preventive measures among Cameroonian.
https://data.worldbank.org/indicator/IT.NET.USER.ZS?locations=CM
Please add the above information to the discussion as a limitation.
Round 2
Reviewer 1 Report
Ok.